# Improved CLIP Training Objective on Fine-Grained Tasks: Tackling False Negatives and Data Noise

## Abstract

Despite its success in various image-text tasks like zero-shot classification on ImageNet, CLIP has been shown to overlook important details in images and captions. This limitation hinders its performance in fine-grained image-text matching tasks. In this paper, we approach this issue through the lens of false negatives (incorrect negative pairs) and data noise (i.e., mislabeled data), which can prevent the model from learning critical details, especially in downstream tasks with a limited number of classes. To address this, we introduce a new loss term incorporating additional supervision to emphasize true negatives. Additionally, we modify the InfoNCE loss to mitigate the impact of data noise. We show that our new method is provably effective under fewer data assumptions than previous approaches, making it particularly suited to noisy multi-modal data. Using the counting task as an example and CLEVR-Count as the benchmark, we demonstrate the performance improvements achieved by our algorithm without requiring extra labeled data.

## 1 INTRODUCTION

As a milestone in multi-modal contrastive learning, CLIP (Radford et al., 2021) effectively learns correspondences between images and text, enabling it to handle a wide range of downstream tasks in a zero-shot manner without prior exposure to labeled examples (Radford et al., 2021). Recently, CLIP has garnered interest on the theoretical side as well (Ren & Li, 2023; Chen et al., 2023; Nakada et al., 2023; Zhang et al., 2023; Xue et al., 2023), with Chen et al. (2023) providing insights into its zero-shot transfer capabilities under certain assumptions about data distributions.

However, recent studies indicate that CLIP performs poorly on *fine-grained* image-text tasks, such as matching spatial relationships (Kamath et al., 2023a), numerical details (Paiss et al., 2023), and compositional semantics (Yuksekgonul et al., 2023; Thrush et al., 2022). For these tasks, CLIP's zero-shot capabilities appear limited. These studies propose several hypotheses and solutions, including fine-tuning CLIP on additional relevant data, hard negative mining, and adding inductive biases to address these issues.

In our work, we approach this problem by examining the effects of false negatives and data noise during CLIP training. First, CLIP treats all non-paired image-text combinations as negative pairs, which can introduce incorrect negative pairs. This issue has been studied in both uni-modal (Huynh et al., 2022b; Chen et al., 2021) and multi-modal contrastive learning contexts (Zolfaghari et al., 2021; Morgado et al., 2021). We observe that for fine-grained tasks with a limited number of classes (e.g., numbers, colors, and directions), the occurrence of false negatives is higher compared to classification tasks with many classes, which can impede the model's ability to learn fine-grained correspondences.

Additionally, using noisy, web-crawled datasets for CLIP training can introduce low-quality positive samples, such as misaligned image-text pairs and ambiguous keywords, further complicating the learning of fine-grained details. For instance, numbers in a caption might refer to dates ("Year two"), counts ("Two cats"), or be irrelevant ("Image Two"). Figure 1 illustrates examples of false negatives and data noise from the DataComp-small dataset (Gadre et al., 2024) in the context of counting: Given the reference image-text pair (a), (b) represents a false negative in the original CLIP training, while (d) is a true negative. Sample (c) is

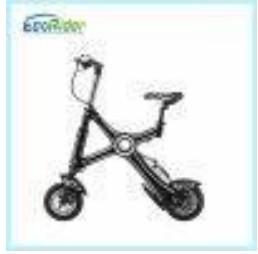

**False Negative**

(b) … **Two** Wheel Electric Bike …

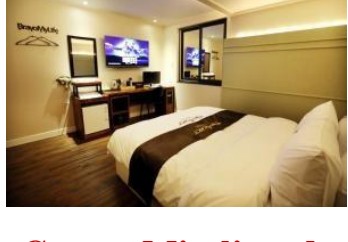

**Counts Misaligned**

(c) Hotel **Two** Heart

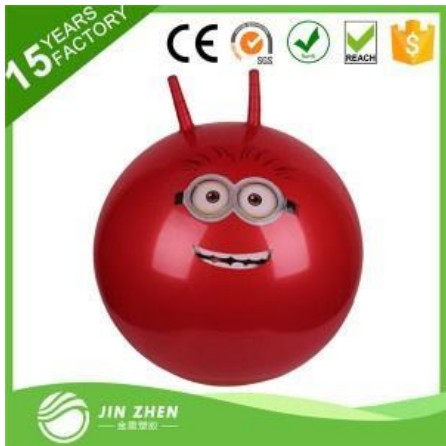

(a) Exercise Hopper Ball with **Two** Sticks for Children

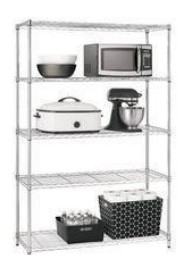

**True Negative**

(d) … **five**-tier shelving unit from Prospace …

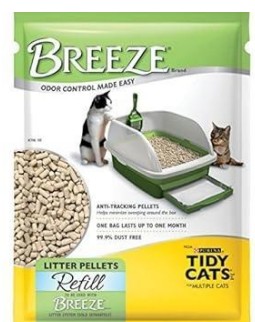

**Irrelevant**

(e) Top Litter & Refills For Cats

Figure 1: Examples in the DataComp-small Dataset

mislabeled as it lacks the concept of "two" in the image, and (e) is an unrelated sample with no numerical information.

**Our Contributions:** We propose a novel multi-modal contrastive training objective to mitigate these issues in CLIP training. Identifying true and false negatives is generally ill-defined but feasible for specific downstream tasks where labels can be obtained through keyword matching. For example, in tasks with labeled image-text pairs, pairs with different labels can be treated as negative samples, allowing us to apply an additional contrastive loss that emphasizes true negatives relevant to the downstream task. Furthermore, we find that modifying the InfoNCE loss could possibly reduce the impact of data noise. Our theoretical analysis derives an upper bound on the top-$r$ loss and explains the effect of this modification using two toy models. Compared to existing analyses of CLIP training (Chen et al., 2023), our approach does not depend on embedding consistency assumptions given shared information, and our framework generalizes to a broader class of contrastive loss objectives.

For our experiments, we use the counting task as an example, comparing our method against the baseline CLIP on the CLEVR-Count benchmark. In this task, we extract numbers from captions as supervised labels and treat image-text pairs with differing numbers as negatives. We observe improved performance when

training with these supervised labels. Additionally, modifying the InfoNCE loss yields performance gains for both the original and supervised CLIP training methods. While our experiments focus on counting, our algorithm is flexible and can be extended to other tasks where supervised labels for image-text pairs are obtainable.

**Organization:** The paper is organized as follows: Section 2 covers related work, Section 3 formally describes the problem setting and introduces the new loss objective, Section 4 provides a theoretical analysis, and Section 5 presents two toy examples. Our experimental results are detailed in Section 6, and we conclude our findings in Section 7.

## 2 RELATED WORK

**Improved CLIP Training for Fine-Grained Tasks** Although OpenAI's CLIP (Radford et al., 2021) and its variants (Ilharco et al., 2021; Zhai et al., 2023; Sun et al., 2023) demonstrate impressive zero-shot classification and retrieval capabilities on popular benchmarks like CIFAR10/100 (Krizhevsky et al., 2009) and ImageNet (Deng et al., 2009; Recht et al., 2019), they have been shown to struggle with fine-grained image-text matching tasks. Such tasks require the model to capture subtle details in both images and text (Thrush et al., 2022; Hsieh et al., 2024; Schiappa et al., 2024; Yuksekgonul et al., 2023; Zhao et al., 2022; Tong et al., 2024a;b; Kamath et al., 2023b). Since CLIP is pretrained on noisy, web-crawled image-text pairs with retrieval as the primary optimization goal, several studies have attributed these challenges to CLIP's dependence on high-quality positive and negative samples. Efforts to address this include hard negative mining for CLIP fine-tuning via keyword replacement (Yuksekgonul et al., 2023; Kamath et al., 2023a) and data cleaning through object detection for tasks like counting (Paiss et al., 2023). However, such approaches rely on external models or data augmentation techniques that may be difficult to generalize to other tasks.

In this work, we explore the potential of leveraging the original training data without constructing new samples, and we provide a theoretical analysis to support this approach. Unlike the detection-based data-cleaning method, our approach does not depend on another model, making it potentially more adaptable to a broader range of downstream tasks.

**Tackling False Negatives in Contrastive Learning** There are many existing works discussing how to tackle false negatives. (Huynh et al., 2022a; Li et al., 2023; 2021; Chun, 2024; Khosla et al., 2021). However, none of them provided theoretical analysis.

In this work, we provided theoretical analysis of our method, with assumption that the correct negative can be eliminated through supervised labels.

**Tackling Data Noises** RINCE (Ching-Yao et al., 2022) provided an alternate loss function to tackle data noises. It regards data noises as a noise distribution added to the correct data.

In this paper, we regard data noises as mislabeled data, so the analysis of RINCE is not applicable.

**Theoretical Analysis of Multi-Modal Contrastive Learning** Previous theoretical studies on multi-modal contrastive learning, particularly as applied to CLIP, have examined its zero-shot transferability (Chen et al., 2023), advantages over uni-modal contrastive learning (Zhang et al., 2023; Nakada et al., 2023), effectiveness with non-isotropic multi-view data (Ren & Li, 2023), and robustness to distribution shifts (Xue et al., 2023).

Our work builds upon the theoretical framework established in Chen et al. (2023), which relies on two key assumptions: (1) Conditional independence, which assumes that the image $x$ and text $y$ are conditionally independent given a discrete, sparse shared feature $z$, and (2) $(\alpha, \beta, \gamma)$-Completeness, which constrains the data distribution by assuming an ideal embedding that, with high probability $(1 - \alpha)$, makes embeddings from different $z$ values distinguishable (by a margin of $\gamma$), while embeddings from the same $z$ do not vary excessively (within a variance $\beta$). We extend this analysis to encompass a broader class of contrastive loss objectives, showing that the $\alpha$ parameter required by different loss functions can vary. Furthermore, we

demonstrate that using supervised labels allows us to relax the constraint on variance ($\beta$), enhancing the flexibility of the theoretical framework.

## 3 PRELIMINARIES

In this section, we introduce the notations and the problem setup, formalize the assumptions on data distribution used in theoretical analysis, and review the original contrastive loss for CLIP training.

**Notations**  Let $[n] = \{1, 2, \cdots, n\}$. We use $\|f\|$ to denote the $L_\infty$ norm of the function $f$. We use the standard $O(\cdot)$ to hide constant factors, and $\tilde{O}(\cdot)$ to hide log factors.

### 3.1 Data Distribution

For data distribution of image $\mathbf{x}$ and text $\mathbf{y}$, we follow the conditional independence assumption in previous work (Chen et al., 2023), but add a new variable $\ell$ to represent the supervised signal. For a relevant data point, its supervised signal is its label in the downstream task. In practice, this supervised signal can be obtained through keyword matching for many tasks. For example, we can use English cardinal numbers as the keywords for the counting task. Note that only a portion of the data has such a supervised signal, and the rest of the data is irrelevant to the target task.

**Assumption 3.1.** *Let $(\mathbf{x}, \mathbf{y}, \ell)$ be generated from the joint distribution $\mathcal{D}_{\mathbf{x} \times \mathbf{y} \times \ell}$. We assume $\mathbf{z}$ to be a shared feature of $\mathbf{x}, \mathbf{y}$ satisfying $\mathbf{x} \perp \mathbf{y} \mid (\mathbf{z}, \ell)$, and $\ell$ is either $\mathbf{z}$ or $\mathbf{0}$. We further denote $(\mathbf{x}, \mathbf{y}, \mathbf{z}, \ell)$ with marginal distributions $\mathcal{D}_{\mathbf{x} \times \mathbf{y} \times \mathbf{z} \times \ell}$.*

*We further assume $\mathbf{z}$ to be a nonzero, discrete and sparse random variable $\mathbf{z} \in \mathcal{V} = \{\mathbf{v}_1, \mathbf{v}_2, \cdots, \mathbf{v}_K\}$ with $p_k := \mathbb{P}[\ell = \mathbf{z} = \mathbf{v}_k], q_k := \mathbb{P}[\ell = \mathbf{0}, \mathbf{z} = \mathbf{v}_k]$, $q = \sum_k q_k$.*

**Remark 3.2.** *$\mathbf{z}$ is the shared feature between image $\mathbf{x}$ and text $\mathbf{y}$, which could be subject names, object counts, or background details. $\ell = \mathbf{0}$ means that the shared feature of the data point is irrelevant to the downstream task, i.e., the test distribution is $\mathcal{D}_{\mathbf{x} \times \mathbf{y} \times \mathbf{z} \mid \ell \neq \mathbf{0}}$.*

### 3.2 Learning via Contrastive Loss

Original CLIP adopts the following loss function

$$L_{S'}(f, \tau) = \frac{-1}{B} \sum_{i \in S'} \log \left( \frac{\exp(f(\mathbf{x}_i, \mathbf{y}_i)/\tau)}{\sum_{j \in S'} \exp(f(\mathbf{x}_i, \mathbf{y}_j)/\tau)} \right) \tag{1}$$

$$- \frac{1}{B} \sum_{i \in S'} \log \left( \frac{\exp(f(\mathbf{x}_i, \mathbf{y}_i)/\tau)}{\sum_{j \in S'} \exp(f(\mathbf{x}_j, \mathbf{y}_i)/\tau)} \right) \tag{2}$$

In practice, there are two encoders $f_{image}$ and $f_{text}$, and $f(\mathbf{x}, \mathbf{y}) = \mathbf{sim}(f_{image}(\mathbf{x}), f_{text}(\mathbf{y}))$. The similarity function $\mathbf{sim}(\cdot, \cdot)$ could be many similarity functions, such as cosine similarity. Below, we use $s(\mathbf{x}_i, \mathbf{y}_j)$ as a shorthand for $\exp(f(\mathbf{x}_i, \mathbf{y}_j)/\tau)$.

## 4 NEW ALGORITHM AND THEORETICAL ANALYSIS

In this section, we first introduce our new algorithm for CLIP training. Then, we prove that this new algorithm does not rely on the consistency assumption in previous analysis (Chen et al., 2023) and explain how the choice of function $g$ affects the bound on top-$r$ error in the downstream task.

### 4.1 Our Loss Function Using Supervised Labels

Our training pipeline is the same as the original CLIP: In each iteration, we sample a batch of image-text pairs $S' = \{\mathbf{x}_i, \mathbf{y}_i, \ell_i\}_{i=1}^B$ from the dataset $S$ and learns a score function $f(\mathbf{x}, \mathbf{y})$ that minimizes a contrastive loss objective on batchs. We only vary the loss function, and the time complexity should remain unchanged. The new loss function is defined as follows:

**Definition 4.1.** *Let $g$ be a concave, monotonically increasing function s.t. $0 \leq g(x) \leq x$ for all $x \geq 0$, we define*

$$L_{g,S'}(f, \tau)$$
$$= \frac{1}{B} \sum_{i \in S', \ell_i \neq \mathbf{0}} g\left( \frac{\sum\limits_{j \in S', \ell_j \neq \mathbf{0}} \mathbb{1}(\ell_j \neq \ell_i) s(\mathbf{x}_i, \mathbf{y}_j)}{s(\mathbf{x}_i, \mathbf{y}_i)} \right).$$

**Remark 4.2.** *Note that when $g = \log(1 + x)$, the loss function is*

$$L_{g,S'}(f, \tau) = \frac{1}{B} \sum_{i \in S', \ell_i \neq \mathbf{0}}$$
$$\log\left( \frac{s(\mathbf{x}_i, \mathbf{y}_i) + \sum\limits_{j \in S', \ell_j \neq \mathbf{0}} \mathbb{1}(\ell_j \neq \ell_i) s(\mathbf{x}_i, \mathbf{y}_j)}{s(\mathbf{x}_i, \mathbf{y}_i)} \right).$$

*This is similar to the first part of the original loss function of CLIP (in (1)), which can be transformed as follows*

$$\frac{-1}{B} \sum_{i \in S'} \log\left( \frac{s(\mathbf{x}_i, \mathbf{y}_i)}{\sum\limits_{j \in S'} s(\mathbf{x}_i, \mathbf{y}_j)} \right)$$
$$= \frac{1}{B} \sum_{i \in S'} \log\left( \frac{s(\mathbf{x}_i, \mathbf{y}_i) + \sum\limits_{j \in S'} \mathbb{1}(j \neq i) s(\mathbf{x}_i, \mathbf{y}_j)}{s(\mathbf{x}_i, \mathbf{y}_i)} \right)$$

Intuitively, $L_{g,S'}(f, \tau)$ is to apply a contrastive loss to the data points relevant to the downstream task with the guarantee that there is no false negative, because samples with the same supervised signals ($\ell_j = \ell_i$) do not appear in the numerator for contrast. Hence, this loss emphasizes the true positives and negative pairs. Note that this loss is asymmetric in that we only apply it to matching images with positive and negative texts in the batch. This is because we extracted the supervised labels only based on texts, so the labels would be much more reliable for picking true negative texts compared with picking the images.

### 4.2 Theoretical Analysis about Correctness

We first show that the empirical loss $\hat{L}_{g,S'}(f, \tau) := (1/N) \sum_{k \in [N]} \hat{L}_{g,S_k}(f, \tau)$ concentrates on the population loss when $N$ is large enough. The population loss of $L_{g,S'}(f, \tau)$ is

$$L_{g,\mathcal{D}^B}(f, \tau)$$
$$= \mathbb{1}(\ell_1 \neq \mathbf{0}) g\left( \frac{\sum\limits_{i \in S', \ell_i \neq \mathbf{0}} \mathbb{1}(\ell_i \neq \ell_1) s(\mathbf{x}_1, \mathbf{y}_i))}{s(\mathbf{x}_1, \mathbf{y}_1)} \right)$$

Similar to the analysis for the original CLIP loss in Chen et al. (2023), we have the following results.

**Theorem 4.3.** *Suppose $\delta \in (0, 1)$, and $g$ and $N$ satisfy*

$$\sup_{x \geq 0} g(x \exp(\epsilon_1)) - g(x) \leq \epsilon/4, and$$
$$N \geq (2\epsilon^{-2} g((B-1) \exp(2M/\tau))) \log(2\mathcal{N}(\mathcal{F}, \epsilon_1 \tau/2)/\delta),$$

*then with probability at least $1 - \delta$, we have*

$$|\hat{L}_{g,S'}(f, \tau) - L_{g,\mathcal{D}^B}(f, \tau)| \leq \epsilon$$

*for all $f \in \mathcal{F}$ and $|f| \leq M$, where $\mathcal{N}(\mathcal{F}, \epsilon)$ is the covering number of $\mathcal{F}$.*

The proof is deferred in the supplementary material. We now state the new assumption for realizability.

**Assumption 4.4** (($\alpha, \gamma$)-Completeness). *There exists a score function $f^*$ bounded by $M$ such that for any $\mathbf{z} \neq \mathbf{z}'$, $\ell$, and $\ell'$, let*

$$\mathbf{x} \sim \mathcal{D}(\mathbf{x} \mid \mathbf{z}, \ell), \mathbf{y} \sim \mathcal{D}(\mathbf{y} \mid \mathbf{z}, \ell)$$
$$\mathbf{x}' \sim \mathcal{D}(\mathbf{x}' \mid \mathbf{z}', \ell'), \mathbf{y}' \sim \mathcal{D}(\mathbf{y}' \mid \mathbf{z}', \ell')$$

*With probability at least $1 - \alpha$, we have $f^*(\mathbf{x}', \mathbf{y}) \leq f^*(\mathbf{x}, \mathbf{y}) - \gamma$ and $f^*(\mathbf{x}, \mathbf{y}') \leq f^*(\mathbf{x}, \mathbf{y}) - \gamma$.*

**Remark 4.5.** *$\gamma$ is the error margin of data with different shared features $\mathbf{z}$. The parameter $\alpha$ allows part of the data to be incorrectly encoded for the best encoding functions. Note that in datasets of web-crawled noisy image-text pairs, $\alpha$ could be quite large due to misaligned image-text pairs.*

This realizability assumption eliminates the consistency assumption in $(\alpha, \beta, \gamma)$-Completeness in the previous work (Chen et al., 2023), which was formalized as

$$\mathbb{E}_{\mathbf{y}, \mathbf{z}}[Var_{\mathbf{x}|\mathbf{z}}(f^*(\mathbf{x}, \mathbf{y}))] \leq \beta,$$
$$\mathbb{E}_{\mathbf{x}, \mathbf{z}}[Var_{\mathbf{y}|\mathbf{z}}(f^*(\mathbf{x}, \mathbf{y}))] \leq \beta.$$

This is to ensure that when false negatives $(\mathbf{x}_j, \mathbf{y}_j)$ for the pair $(\mathbf{x}_i, \mathbf{y}_i)$ appear in the same batch, the score function would not assign a much larger score to the pairs $(\mathbf{x}_i, \mathbf{y}_j)$ and $(\mathbf{x}_j, \mathbf{y}_i)$, which is not favored by the contrastive learning mechanism. This is possible, especially when the unique features count a lot for the resulting encoding. Our algorithm circumvents this issue by restricting the comparison between positives and true negative samples in the loss term.

Now we present the theoretical guarantee for the "good score function" $f^*$ achieving low training loss.

**Lemma 4.6.** *Let $f^*$ be the function satisfies **Assumption 4.4**, we have:*

$$\mathbb{E}[L_{g,\mathcal{D}^B}(f^*, \tau)] \leq q^2 B(\alpha g(\exp(2M/\tau)) + \exp(-\gamma/\tau))$$

*Proof of Lemma 4.6.* Let the event $\mathcal{E}_t$ be the case that $z_t \neq z_1$ and $f^*(x_1, y_t) - f^*(x_1, y_1) \leq -\gamma$, then by $(\alpha, \gamma)$-assumption we have $\Pr[\ell \neq \mathbf{0}, \mathcal{E}_t^c | \mathbf{z}_1, \ell_1, \mathbf{z}_t, \ell_t] \leq q\alpha$ for all $(\mathbf{z}_1, \ell_1, \mathbf{z}_t, \ell_t)$.

$$
\begin{aligned}
&L_{g, \mathcal{D}^B}(f^*, \tau) \\
&= \mathbb{E}[\mathbb{1}(\ell_1 \neq \mathbf{0}) g(\sum_{t \in [B]} \mathbb{1}(\ell_t \neq \mathbf{0}, \mathbf{z}_t \neq \mathbf{z}_1, \mathcal{E}_t^c) \\
&\qquad\qquad \exp((f^*(\mathbf{x}_1, \mathbf{y}_t) - f^*(\mathbf{x}_1, \mathbf{y}_1))/\tau) \\
&\quad + \sum_{t \in [B]} \mathbb{1}(\ell_t \neq \mathbf{0}, \mathbf{z}_t \neq \mathbf{z}_1, \mathcal{E}_t) \\
&\qquad\qquad \exp((f^*(\mathbf{x}_1, \mathbf{y}_t) - f^*(\mathbf{x}_1, \mathbf{y}_1))/\tau))] \\
&\leq \mathbb{E}[\mathbb{1}(\ell_1 \neq \mathbf{0}) g(\sum_{t \in [B]} \mathbb{1}(\ell_t \neq \mathbf{0}, \mathbf{z}_t \neq \mathbf{z}_1, \mathcal{E}_t^c) \\
&\qquad\qquad \exp((f^*(\mathbf{x}_1, \mathbf{y}_t) - f^*(\mathbf{x}_1, \mathbf{y}_1))/\tau) \\
&\quad + \sum_{t \in [B]} \mathbb{1}(\ell_t \neq \mathbf{0}, \mathbf{z}_t \neq \mathbf{z}_1, \mathcal{E}_t) \\
&\qquad\qquad \exp((f^*(\mathbf{x}_1, \mathbf{y}_t) - f^*(\mathbf{x}_1, \mathbf{y}_1))/\tau))] \\
&\leq \mathbb{E}[\mathbb{1}(\ell_1 \neq \mathbf{0}) g(\sum_{t \in [B]} \mathbb{1}(\ell_t \neq \mathbf{0}, \mathbf{z}_t \neq \mathbf{z}_1, \mathcal{E}_t^c) \exp(2M/\tau) \\
&\quad + \sum_{t \in [B]} \mathbb{1}(\ell_t \neq \mathbf{0}, \mathbf{z}_t \neq \mathbf{z}_1, \mathcal{E}_t) \exp(-\gamma/\tau))] \\
&\leq \mathbb{E}[\mathbb{1}(\ell_1 \neq \mathbf{0})(\sum_{t \in [B]} \mathbb{1}(\ell_t \neq \mathbf{0}, \mathbf{z}_t \neq \mathbf{z}_1, \mathcal{E}_t^c) g(\exp(2M/\tau)) \\
&\quad + \sum_{t \in [B]} \mathbb{1}(\ell_t \neq \mathbf{0}, \mathbf{z}_t \neq \mathbf{z}_1, \mathcal{E}_t) \exp(-\gamma/\tau))] \\
&\leq q^2 B(\alpha g(\exp(2M/\tau)) + \exp(-\gamma/\tau))
\end{aligned}
$$

$\square$

The key observation in the proof is that the "bad" event only happens for mistaking the positive and the true negatives.

Combined with **Theorem 4.3**, this leads to our main result below:

**Theorem 4.7.** *Suppose **Assumption 4.4** hold and we can find an $\epsilon$-approximate $\widehat{f} \in \mathcal{F}$ with respect to the temperature $\tau$ such that $\widehat{f}$ is bounded by $M$ and $L_{g, \mathcal{D}^B}(\widehat{f}, \tau) \leq L_{g, \mathcal{D}^B}(f^*, \tau) + \epsilon$, then:*

*For $(\mathbf{x}, \mathbf{z}) \sim \mathcal{D}_{\mathbf{x} \times \mathbf{z} | \ell = \mathbf{z}}$, $\{\mathbf{y}_k \sim \mathcal{D}_{\mathbf{y} | \ell = v_k}\}$, , let $\mathbf{y}^* = \sum_{k \in [K]} \mathbb{1}(\mathbf{z} = \mathbf{v_k}) \mathbf{y}_k$, we have:*

$$
\mathbb{E}[g(-1 + \sum_{k \in [K]} \exp((\widehat{f}(\mathbf{x}, \mathbf{y}_t) - \widehat{f}(\mathbf{x}, \mathbf{y}^*))/\tau))] \leq \epsilon'
$$

*where $\epsilon' = (C_B + 2) \cdot (qB(\alpha g(\exp(2M/\tau)) + \exp(-\gamma/\tau)) + q^{-1}\epsilon)$, and $C_B = \widetilde{O}(\max_k p_k^{-1}/B)$.*

*Proof of Theorem 4.5.* First, by **Lemma 4.6**, we have

$$
L_{g, \mathcal{D}^B}(\widehat{f}, \tau) \leq \epsilon + q^2 B(\alpha g(\exp(2M/\tau)) + \exp(-\gamma/\tau))
$$

Notice that:

$$L_{g,\mathcal{D}^B}(\widehat{f},\tau)$$

$$=\mathbb{E}[\mathbb{1}(\ell_1 \neq \mathbf{0})g(\sum_{t\in[B]} \mathbb{1}(\ell_t \neq \mathbf{0}, \mathbf{z}_t \neq \mathbf{z}_1)$$

$$\exp((\widehat{f}(\mathbf{x}_1,\mathbf{y}_t) - \widehat{f}(\mathbf{x}_1,\mathbf{y}_1))/\tau)]$$

$$= \Pr[\mathbb{1}(\ell_1 \neq \mathbf{0})]\mathbb{E}[g(\sum_{t\in[B]} \mathbb{1}(\ell_t \neq \mathbf{0}, \mathbf{z}_t \neq \mathbf{z}_1)$$

$$\exp((\widehat{f}(\mathbf{x}_1,\mathbf{y}_t) - \widehat{f}(\mathbf{x}_1,\mathbf{y}_1))/\tau)|\ell_1 \neq \mathbf{0}]$$

$$=q\mathbb{E}[g(\sum_{t\in[B]} \mathbb{1}(\ell_t \neq \mathbf{0}, \mathbf{z}_t \neq \mathbf{z}_1)$$

$$\exp((\widehat{f}(\mathbf{x}_1,\mathbf{y}_t) - \widehat{f}(\mathbf{x}_1,\mathbf{y}_1))/\tau)|\ell_1 \neq \mathbf{0}]$$

So, we have

$$\mathbb{E}[g(\sum_{t\in[B]} \mathbb{1}(\ell_t \neq \mathbf{0}, \mathbf{z}_t \neq \mathbf{z}_1)$$

$$\exp((\widehat{f}(\mathbf{x}_1,\mathbf{y}_t) - \widehat{f}(\mathbf{x}_1,\mathbf{y}_1))/\tau)|\ell_1 \neq \mathbf{0}]$$

$$\leq q^{-1}\epsilon + qB(\alpha g(\exp(2M/\tau)) + \exp(-\gamma/\tau))$$

We generate sequences $(\mathbf{z}_1',\ell_1'),(\mathbf{z}_2',\ell_2'),\cdots,(\mathbf{z}_L',\ell_L')$, with $L = \lceil\log(2K)/\min p_k\rceil$, then with probability at least $\frac{1}{2}$ it will cover all $v_k$.

Then, we introduce $\lceil L/(B-1)\rceil$ copies of $(\mathbf{z}_2^{(l)},\ell_2^{(l)}),(\mathbf{z}_3^{(l)},\ell_3^{(l)}),\cdots,(\mathbf{z}_B^{(l)},\ell_B^{(l)})$, then by concavity and monotonicity of $g$, we have

$$\lceil L/(B-1)\rceil\mathbb{E}[g(\sum_{t\in[B]} \mathbb{1}(\ell_t \neq \mathbf{0}, \mathbf{z}_t \neq \mathbf{z}_1)$$

$$\exp((\widehat{f}(\mathbf{x}_1,\mathbf{y}_t) - \widehat{f}(\mathbf{x}_1,\mathbf{y}_1))/\tau)|\ell_1 \neq \mathbf{0}, z_1]$$

$$\geq\mathbb{E}[g(\sum_{t\in[L]} \mathbb{1}(\ell_t \neq \mathbf{0}, \mathbf{z}_t' \neq \mathbf{z}_1)$$

$$\exp((\widehat{f}(\mathbf{x}_1,\mathbf{y}_t') - \widehat{f}(\mathbf{x}_1,\mathbf{y}_1))/\tau)|\ell_1 \neq \mathbf{0}, z_1]$$

$$\geq\frac{1}{2}\mathbb{E}[g(-1 + \sum_{k\in[K]} \exp((\widehat{f}(\mathbf{x},\mathbf{y}_t) - \widehat{f}(\mathbf{x},\mathbf{y}^*))/\tau))|z]$$

Let $C_B = 2\lceil L/(B-1)\rceil$, we complete the proof. $\qquad\square$

This result can be translated into the bound on the top-$r$ error of the downstream task: We calculate the image-text score for a given image $\mathbf{x}$ and a set of texts $\mathbf{y}_k$ for all $k \in [K]$, and then pick the top-$r$ highest score to check if the correct text is included.

**Corollary 4.8.** *Suppose the results of **Theorem 4.7** hold for the learned function $\widehat{f}$, then the top-r error is bounded by $\epsilon'/g(r)$.*

We can see that the theoretical guarantee of our new loss objective is not constrained by $\beta$ since $\epsilon'$ is not related to $\beta$. Hence, we prove that through additional comparison between positives and true negatives, the consistency assumption on image-text pairs with the same supervised label is relaxed.

On the other hand, our theoretical results hold for function $g$ beyond the original choice $(\log(1 + x))$. We note that for different available functions $g$, the theoretical guarantee on the top-$r$ error varies.

**Remark 4.9** (Choice of Function $g$). *If $\alpha > 0$, choosing a better function g will probably result in a smaller $g(\exp(2M/\tau))/g(r)$, and consequently, a smaller top-r error.*

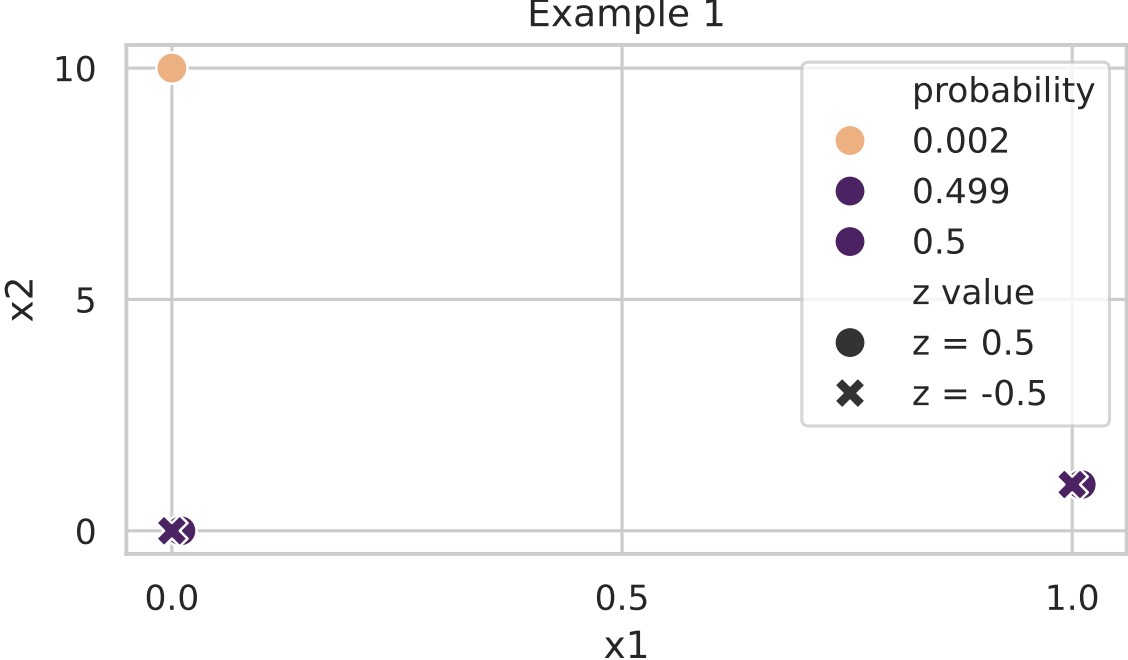

Figure 2: Illustration of Example 1

When $\alpha$ is large, e.g., in a noisy dataset, this effect is more prominent. In practice, researchers can determine the best choice of $g$ by performance on a validation set. We demonstrate the effect of the function choice in the experiment part.

## 5 CASE STUDY

In this section, we demonstrate the effect of the function $g$'s choice and the comparison of positive and true negatives with two specific examples.

### 5.1 Choice of Function $g$

We mainly focus on two specific cases of function $g$: $g_1(x) = \log(1 + x)$, $g_2(x) = \frac{x}{1+x}$.

Since $g_2(exp(2M/\tau)) \leq 1$, $g_1(exp(2M/\tau)) \geq 2M/\tau$. we expect that $g_2$ performs better than $g_1$ when $\alpha > 0$, and $2M/\tau$ is large enough. Intuitively, the incorrect data in the dataset will have less influence on the loss function with $g_2(x) = \frac{x}{1+x}$.

The following example shows a situation where using $g_2 = \frac{x}{1+x}$ is better than using $g_1 = \log(1 + x)$. An illustration of this example is in Figure 2.

**Definition 5.1** (Example 1). *Let $\ell = \mathbf{z} \in \mathbb{R}^1$ be random variable uniformly drawn from $\mathcal{V} = \{\mathbf{v}_1 = 0.5, \mathbf{v}_2 = -0.5\}$. Set $\mathbf{y} = \mathbf{z}$. And $\mathbf{x} \in \mathbb{R}^2$ is generated with the following probability:*

$$\Pr[\mathbf{x} = (1.01, 1) \mid \mathbf{z} = 0.5]$$
$$= \Pr[\mathbf{x} = (0.01, 0) \mid \mathbf{z} = 0.5] = 0.499,$$
$$\Pr[\mathbf{x} = (0, 10) \mid \mathbf{z} = 0.5] = 0.002,$$
$$\Pr[\mathbf{x} = (1, 1) \mid \mathbf{z} = -0.5]$$
$$= \Pr[\mathbf{x} = (0, 0) \mid \mathbf{z} = -0.5] = 0.5$$

*The hypothesis space is $f_{a,b,c}(\mathbf{x} = (x_1, x_2), \mathbf{y}) = y(a \cdot x_1 + b \cdot x_2 + c)$.*

We can verify that $f^* = f_{0.1,-0.1,-0.0005}$ satisfies $(\alpha, \gamma)$-Completeness with $M = 1, \alpha = 0.002, \gamma = 0.0005$.

So, by **Theorem 4.7**, we have:

$$\epsilon' = (C_B + 2) \cdot (B(0.002g(\exp(2/\tau)) \\ + \exp(-0.0005/\tau)) + \epsilon)$$

If $B = 2$, $g = g_2$, $\tau = 0.00005$, $\epsilon < 0.001$, we have $C_B \leq 6$:

$$\begin{aligned}
\epsilon' &= (C_B + 2) \cdot (B(0.002g(\exp(2/\tau)) \\
&\quad + \exp(-0.0005/\tau)) + \epsilon) \\
&\leq 8 \cdot (2 \cdot (0.002g(\exp(2/\tau)) + \exp(-10)) + 0.001) \\
&\leq 8 \cdot (2 \cdot (0.002 + \exp(-10)) + 0.001) \\
&< 0.05
\end{aligned}$$

the second inequality comes from the fact that $g_2(x) < 1$ for all $g$.

By **Corollary 4.8**, the top-1 error of $\widehat{f}$ is bounded by $\epsilon'/g(1) < 0.05/0.5 = 0.1$.

However, if we use $g = g_1$, we would probably get $a \approx 3.7 \times 10^{-6}, b \approx -5.6 \times 10^{-6}, c \approx 0.8 \times 10^{-7}$, which has the minimum loss, but the top-1 error is 0.4995.

The intuitive understanding of this example is that the data $((0, 10), 0.5, 0.5)$ is incorrect, and it has a larger influence on the loss function with $g = g_1$. In the noisy web-crawled dataset, the incorrectly-paired image-text data appear very often and can have a heavy negative impact on training. Previously, a common approach was to perform data filtering before training, but this relies on building heuristics about "good data." We suggest that changing the function in the contrastive loss objective could possibly tackle this issue from a different perspective.

## 5.2 Effect of Additional Supervision

Since we change the original assumption $(\alpha, \beta, \gamma)$-Completeness to the assumption $(\alpha, \gamma)$-Completeness in our proof, we expect that the additional supervision is helpful when the good score function satisfies the $(\alpha, \gamma)$-Completeness, but not the $(\alpha, \beta, \gamma)$-Completeness. In other words, since the original model tend to get a function with a small $\mathbb{E}_{y,z}[Var_{x|z}(f^*(x,y))]$ and $\mathbb{E}_{x,z}[Var_{y|z}(f^*(x,y))]$, if we have a problem setting that similarity functions with small $\mathbb{E}_{y,z}[Var_{x|z}(f^*(x,y))]$ and $\mathbb{E}_{x,z}[Var_{y|z}(f^*(x,y))]$ have bad performances, we can expect that our new method leads to better models.

The following example shows that the new loss adapts better than the original CLIP loss in this case. An illustration of the example is shown in Figure 3.

**Definition 5.2** (Example 2). *Let $\ell = \mathbf{z} \in \mathbb{R}^1$ be random variable drawn from $\mathcal{V} = \{\mathbf{v}_1 = 1, \mathbf{v}_2 = -1\}$, with $\Pr[\mathbf{z} = 1] = p_1, \Pr[\mathbf{z} = -1] = p_2, p_1 + p_2 = 1, p_2 < \frac{1}{4}$. Set $\mathbf{x} = \mathbf{z}$. And $\mathbf{y} \in [-\pi, \pi]$ is generated with the following probability:*

$$\Pr[\mathbf{y} = -\theta_0 \mid \mathbf{z} = 1] = \Pr[\mathbf{y} = \theta_0 \mid \mathbf{z} = 1] = \frac{p_1 - p_2}{2p_1},$$

$$\Pr[\mathbf{y} = -5\theta_0 \mid \mathbf{z} = 1] = \frac{p_2}{p_1},$$

$$\Pr[\mathbf{y} = 3\theta_0 \mid \mathbf{z} = -1] = 1$$

*where $\theta_0$ is a parameter satisfying $0 < \theta_0 < \frac{\pi}{10}$.*

*The hypothesis space is $f_\theta(\mathbf{x}, \mathbf{y}) = \mathbf{x}\cos(y + \theta)$ for $\theta \in [-\pi, \pi)$.*

Then, if $-\theta_0 < \theta < \theta_0$, $\theta < -\pi + \theta_0$ or $\theta > \pi - \theta_0$, then the top-1 error is at least $p_2$.

Let $B = 2$, the following theorem shows that the performance of the original CLIP is not satisfying:

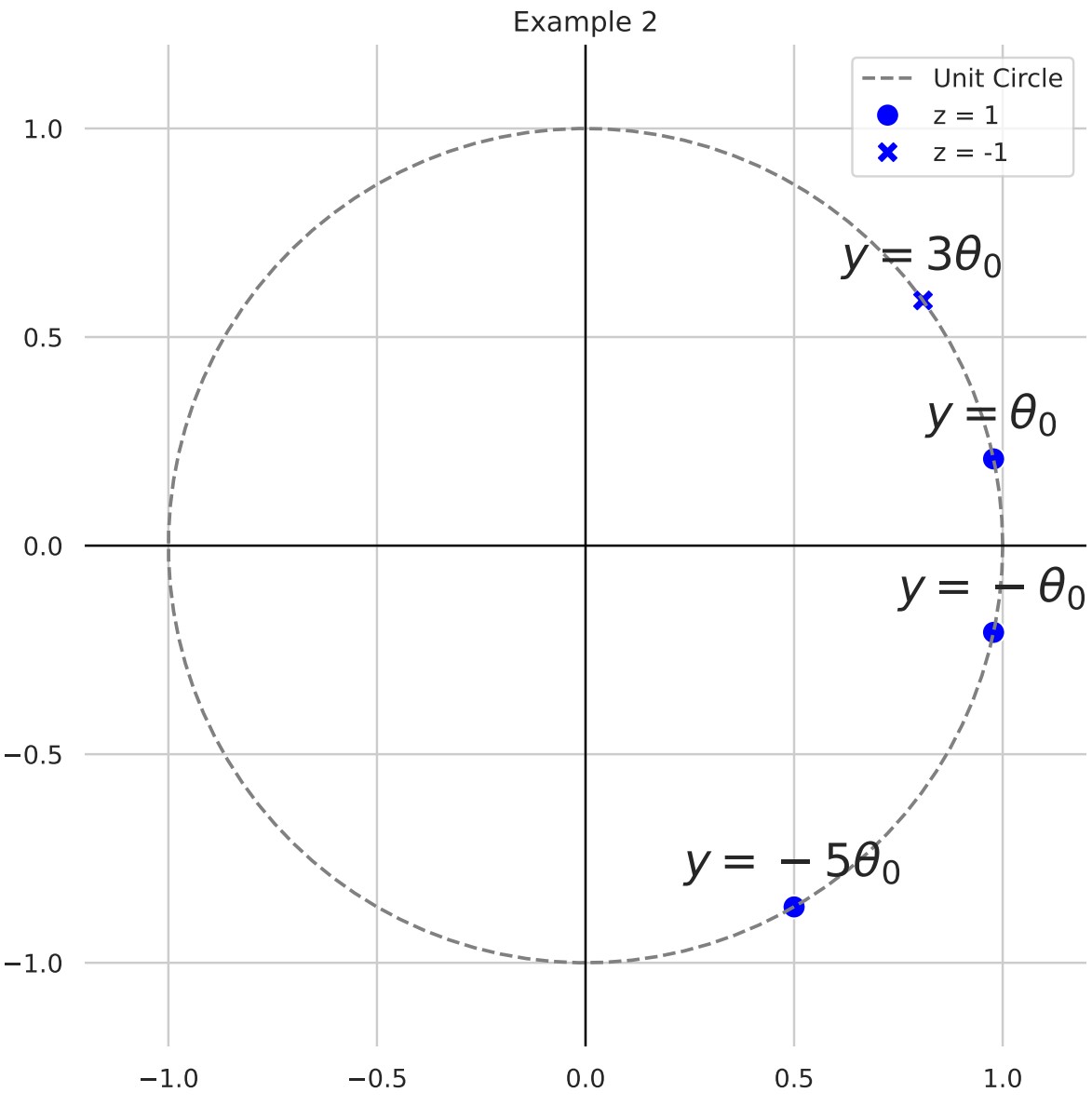

Figure 3: Illustration of Example 2 $(\theta_0 = \frac{\pi}{15})$

**Theorem 5.3.** *there is a constant $\delta'$ and a constant $C'$ such that:*

*For any $p_2 < \delta$, $0 < \tau < 1$, $\theta \in [-\pi + \theta_0, -\theta_0] \cup [\theta_0, \pi - \theta_0]$ we have:*

$$L_{\mathcal{D}^B}(f_\theta, \tau) \geq L_{\mathcal{D}^B}(f_0, \tau) + C'/\tau$$

The theorem shows that if $L_{\mathcal{D}^B}(\widehat{f}, \tau) < L_{\mathcal{D}^B}(f_0, \tau) + C'/\tau$, then $\theta \in [-\pi, -\pi + \theta_0) \cup (-\theta_0, \theta_0) \cup (\pi - \theta_0, \pi)$, and thus the top-1 error is at least $p_2$. The proof is deferred to the supplementary material.

In other words, the theorem shows that all "good" similarity functions have large losses. So, by minimizing the loss, we would probably get a bad similarity function.

However, if we use our loss function, then $f^* = f_{\frac{\pi}{2} - \frac{5}{2}\theta_0}$, satisfies $(\alpha, \gamma)$-Completeness with $\alpha = 0, \gamma = 2\sin(\frac{\theta_0}{2})$, though with a large $\beta$.

Table 1: Accuracies for Varied Methods

| $\eta$ | $g_1(x) = \log(1+x)$ | $g_2(x) = \frac{x}{1+x}$ |
|---|---|---|
| 0 (baseline) | 0.1405($\pm$0.0051) | |
| 100 | 0.1448($\pm$0.25) | 0.1401($\pm$0.026) |
| 1000 | 0.1669($\pm$0.021) | 0.1575($\pm$0.002) |

Table 2: Accuracies of an Alternative Loss Function

| $\eta$ | $g_1(x) = \log(1+x)$ | $g_2(x) = \frac{x}{1+x}$ |
|---|---|---|
| 0 (baseline) | 0.1405($\pm$0.0051) | |
| 100 | 0.1508($\pm$0.046) | 0.1385($\pm$0.022) |
| 1000 | 0.1367($\pm$0.010) | 0.1481($\pm$0.018) |

So, by **Theorem 4.7**, we have:

$$\epsilon' = (C_B + 2) \cdot (B \exp(-\gamma/\tau) + \epsilon)$$
$$= O(\exp(-\gamma/\tau) + \epsilon)$$

which will be less than $p_2 g(2)$ when $\tau$ and $\epsilon$ are small enough. By **Corollary 4.8**, the top-1 loss will be less than $p_2$.

## 6 EXPERIMENT

In this section, we conduct experiments on using our new loss for CLIP's pretraining. We show that emphasizing the contrastive loss between positives and true negatives benefits the corresponding downstream task. We further discuss how the choice of function $g$ and the denominator affects training as ablation studies.

**Experiment Settings**  We train the ViT-B/32 models by minimizing the loss $L_{g,S'}(f, \eta, \tau) = L_{S'}(f, \tau) + \eta L_{g,S'}(f, \tau)$. We primarily focus on DataComp-small as the training dataset with batch size $B = 4096$, temperature parameter $\tau = 0.07$ learning rate$= 5 \times 10^{-4}$, warmup$= 500$. Our choice of hyperparameters follows the setting in the DataComp-small challenge (Gadre et al., 2024). To obtain the supervised signal, we use English cardinal numbers from "two" to "twenty" (e.g, "two", "three", "four", "five", ...) as the keywords and perform keyword matching on captions to find whether one of a word in the text caption is a keyword. We use CLEVR-Count as the benchmark and evaluate the top-1 correctness. We trained the baseline twice and each variation of our methods three times on a single A40 GPU, and computed the average loss along with standard error.

**Additional supervised labels bring improvements over baseline CLIP.**  Table 1 shows the top-1 accuracy on the CLEVR-Count benchmark for varied methods. We observe that using the additional loss term ($\eta \neq 0$) with either choice of $g$ helps the model perform better than the baseline in the counting task on average. This implies that in the baseline training setting, the numerical comparison between samples was not fully utilized. Additionally, it verifies that our supervised labels for picking positives and true negatives are simple yet effective. We note that the keyword matching strategy can be further applied to obtaining similar fine-grained supervised labels like directions and colors.

**Using false negatives in the denominator.**  Another possible loss term to better utilize the false negatives is to use them as positive samples. We try this idea in CLIP's pretraining too. Instead of using

the loss function

$$
L_{g,S'}(f,\tau)
$$
$$
= \frac{1}{B} \sum_{i \in S', \ell_i \neq \mathbf{0}} g \left( \frac{\sum\limits_{j \in S', \ell_j \neq \mathbf{0}} \mathbb{1}(\ell_j \neq \ell_i) s(\mathbf{x}_i, \mathbf{y}_j)}{s(\mathbf{x}_i, \mathbf{y}_i)} \right),
$$

we use

$$
L'_{g,S'}(f,\tau)
$$
$$
= \frac{1}{B} \sum_{i \in S', \ell_i \neq \mathbf{0}} g \left( \frac{\sum\limits_{j \in S', \ell_j \neq \mathbf{0}} \mathbb{1}(\ell_j \neq \ell_i) s(\mathbf{x}_i, \mathbf{y}_j)}{\sum\limits_{j \in S', \ell_j \neq \mathbf{0}} \mathbb{1}(\ell_j = \ell_i) s(\mathbf{x}_i, \mathbf{y}_j)} \right).
$$

(Actually it is the method $\mathcal{L}_{in}^{sup}$ shown in Khosla et al. (2021))

We find that this strategy yields stronger models than the baseline as well but is less effective than our loss term. The possible reason could be that the false negatives differ from the positives in aspects other than numbers, such as the subject type. Hence, forcing the alignment of their embeddings might undermine the model's ability to form holistic representations. This suggests that the counting task is not separated from other perception and visual reasoning tasks.

**Choice of Function** $g$.  We test two different function $g$ in the additional loss term: $g_1(x) = \log(1 + x)$ (used in original CLIP loss) and $g_2(x) = \frac{x}{1+x}$. The results are shown in the two columns of Table 1 and Table 2. The performance of $g_2(x) = \frac{x}{1+x}$ is not as good as we expect. Maybe this loss function is harder to train than $g_1(x) = \log(1 + x)$. However, in $\eta = 1000$ case, we find that the $g_2(x) = \frac{x}{1+x}$ model's performance is more stable.

**Performance on other benchmarks.**  We also tested our model on other common benchmarks for CLIP evaluation. The results are detailed in Appendix C. Since these tasks are not about counting, we cannot expect the performance to be better than the baseline. We found that the performance is not worsened too much.

## 7 CONCLUSION

In this work, we propose a novel contrastive loss objective designed to enhance performance on downstream tasks involving image-text pairs with supervised labels. Our theoretical analysis suggests that this objective can improve model robustness by reducing the impact of false negatives and mitigating data noise. Empirically, we validate the effectiveness of our approach through experiments on the counting benchmark, demonstrating notable improvements. We hope this work inspires further exploration into optimizing CLIP training with noisy image-text data for specific downstream tasks.

**Limitations and future work** Our current experiments are primarily conducted using the Datacomp-small training dataset and evaluated on the CLEVR-Count test dataset. Future work should extend this analysis to a broader range of datasets to better understand the generalizability of our approach. Additionally, we have only explored the new loss objective in the context of CLIP pretraining. It would be valuable to investigate its potential in CLIP finetuning, especially for adapting the model to specific tasks. Another promising direction for future research is to explore the application of this contrastive loss framework to other multi-modal learning models, potentially broadening its impact across various domains and tasks.

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

## A MISSING PROOFS

### A.1 Proof of Results in Section 4

In this section, we present the detailed proof of results in Section 4.

*Proof of Theorem 4.3.* First, we have

$$
L_{g,S'}(f,\tau) = \frac{1}{B} \sum_{i \in S'} \mathbb{1}(\ell_i \neq \mathbf{0}) g \left( \sum_{j \in S'} \mathbb{1}(\ell_j \neq \mathbf{0}, \ell_j \neq \ell_i) \exp((f(x_i, y_j) - f(x_i, y_i))/\tau) \right)
$$

$$
\leq \frac{1}{B} \sum_{i \in S'} g \left( \sum_{j \in S', j \neq i} \exp(2M/\tau) \right)
$$

$$
= \frac{1}{B} \sum_{i \in S'} g \left( (B-1) \exp(2M/\tau) \right)
$$

$$
= g \left( (B-1) \exp(2M/\tau) \right)
$$

Then, for any $\|f_1 - f_2\| \leq \epsilon_1 \tau/2$, we have

$$
L_{g,S'}(f_1, \tau) - L_{g,S'}(f_2, \tau) = \frac{1}{B} \sum_{i \in S'} \mathbb{1}(\ell_i \neq \mathbf{0}) g \left( \sum_{j \in S'} \mathbb{1}(\ell_j \neq \mathbf{0}, \ell_j \neq \ell_i) \exp((f_1(x_i, y_j) - f)_1(x_i, y_i))/\tau) \right)
$$

$$
- \frac{1}{B} \sum_{i \in S'} \mathbb{1}(\ell_i \neq \mathbf{0}) g \left( \sum_{j \in S'} \mathbb{1}(\ell_j \neq \mathbf{0}, \ell_j \neq \ell_i) \exp((f_2(x_i, y_j) - f_2(x_i, y_i))/\tau) \right)
$$

$$
\leq \frac{1}{B} \sum_{i \in S'} \mathbb{1}(\ell_i \neq \mathbf{0}) g \left( \sum_{j \in S'} \mathbb{1}(\ell_j \neq \mathbf{0}, \ell_j \neq \ell_i) \exp((f_1(x_i, y_j) - f)_1(x_i, y_i))/\tau) \right)
$$

$$
- \frac{1}{B} \sum_{i \in S'} \mathbb{1}(\ell_i \neq \mathbf{0}) g \left( \sum_{j \in S'} \mathbb{1}(\ell_j \neq \mathbf{0}, \ell_j \neq \ell_i) \exp((f_1(x_i, y_j) - f_1(x_i, y_i) - \epsilon_1\tau)/\tau) \right)
$$

$$
= \frac{1}{B} \sum_{i \in S'} \mathbb{1}(\ell_i \neq \mathbf{0}) g \left( \sum_{j \in S'} \mathbb{1}(\ell_j \neq \mathbf{0}, \ell_j \neq \ell_i) \exp((f_1(x_i, y_j) - f)_1(x_i, y_i))/\tau) \right)
$$

$$
- \frac{1}{B} \sum_{i \in S'} \mathbb{1}(\ell_i \neq \mathbf{0}) g \left( \exp(-\epsilon_1) \sum_{j \in S'} \mathbb{1}(\ell_j \neq \mathbf{0}, \ell_j \neq \ell_i) \exp((f_1(x_i, y_j) - f_1(x_i, y_i))/\tau) \right)
$$

$$
\leq \epsilon/2
$$

By definition of covering number, we can cover $\mathcal{F}$ by $K = \mathcal{N}(\mathcal{F}, \epsilon_1\tau/2)$ subsets $\mathcal{B}_1, \mathcal{B}_2, \cdots, \mathcal{B}_K$, where $\mathcal{B}_1, \mathcal{B}_2, \cdots, \mathcal{B}_K$ are balls with radius $\epsilon_1\tau/2$ centered at $f_1, f_2, \cdots, f_K$. So, we have

$$
\Pr \left[ \sup_{f \in \mathcal{F}} \left| \hat{L}_{g,S'}(f, \tau) - L_{g,\mathcal{D}^B}(f, \tau) \right| \geq \epsilon \right] \leq \sum_{k \in [K]} \Pr \left[ \sup_{f \in \mathcal{B}_k} \left| \hat{L}_{g,S'}(f, \tau) - L_{g,\mathcal{D}^B}(f, \tau) \right| \geq \epsilon \right]
$$

$$
\leq \sum_{k \in [K]} \Pr \left[ \left| \hat{L}_{g,S'}(f_k, \tau) - L_{g,\mathcal{D}^B}(f_k, \tau) \right| \geq \epsilon/2 \right]
$$

$$
\leq 2K \exp \left( \frac{N\epsilon^2}{2g \left( (B-1) \exp(2M/\tau) \right)} \right)
$$

With $K = \mathcal{N}(\mathcal{F}, \epsilon_1\tau/2)$ and $N \geq (2\epsilon^{-2}g((B-1)\exp(2M/\tau)))\log(2\mathcal{N}(\mathcal{F}, \epsilon_1\tau/2)/\delta)$, we complete the proof. $\square$

*Proof of Corollary 4.8.* Let $\mathcal{E}$ be the event that all the top-$r$ choices give the wrong prediction.
By **Lemma 4.6**, we have

$$
\begin{aligned}
\epsilon' &\geq \mathbb{E}[g(-1 + \sum_{k\in[K]} \exp((\widehat{f}(\mathbf{x}, \mathbf{y}_t) - \widehat{f}(\mathbf{x}, \mathbf{y}^*))/\tau))] \\
&\geq \mathbb{E}[g(-1 + \sum_{k\in[K]} \mathbb{1}[\widehat{f}(\mathbf{x}, \mathbf{y}_t) - \widehat{f}(\mathbf{x}, \mathbf{y}^*) \geq 0])] \\
&\geq g(r)\Pr[\sum_{k\in[K]} \mathbb{1}[\widehat{f}(\mathbf{x}, \mathbf{y}_t) - \widehat{f}(\mathbf{x}, \mathbf{y}^*) \geq 0] \geq 1+r] \\
&\geq g(r)\Pr[\mathcal{E}]
\end{aligned}
$$

$\square$

## A.2   Proof of Results in Section 5

In this section, we present the detailed proof of results in Section 5.

*Proof of Theorem 5.3.* Let $\mathcal{E}$ be the event that $\mathbf{x}_1 = \mathbf{x}_2 = 1, \mathbf{y}_1, \mathbf{y}_2 \in \{-\theta_0, \theta_0\}$. Then for any $f$, We have

$$
\begin{aligned}
L_{\mathcal{D}^B}(f, \tau) &= \mathbb{E}[\log(1 + \exp((f(\mathbf{x}_1, \mathbf{y}_2) - f(\mathbf{x}_1, \mathbf{y}_1))/\tau)) + \log(1 + \exp((f(\mathbf{x}_2, \mathbf{y}_1) - f(\mathbf{x}_1, \mathbf{y}_1))/\tau))] \\
&= \Pr[\mathcal{E}]\mathbb{E}[\log(1 + \exp((f(\mathbf{x}_1, \mathbf{y}_2) - f(\mathbf{x}_1, \mathbf{y}_1))/\tau)) + \log(1 + \exp((f(\mathbf{x}_2, \mathbf{y}_1) - f(\mathbf{x}_1, \mathbf{y}_1))/\tau)) | \mathcal{E}] \\
&\quad + \Pr[\mathcal{E}^c]\mathbb{E}[\log(1 + \exp((f(\mathbf{x}_1, \mathbf{y}_2) - f(\mathbf{x}_1, \mathbf{y}_1))/\tau)) + \log(1 + \exp((f(\mathbf{x}_2, \mathbf{y}_1) - f(\mathbf{x}_1, \mathbf{y}_1))/\tau)) | \mathcal{E}^c] \\
&= \Pr[\mathcal{E}]\mathbb{E}[\log(1 + \exp((f(\mathbf{x}_1, \mathbf{y}_2) - f(\mathbf{x}_1, \mathbf{y}_1))/\tau)) + \log 2 | \mathcal{E}] \\
&\quad + \Pr[\mathcal{E}^c]\mathbb{E}[\log(1 + \exp((f(\mathbf{x}_1, \mathbf{y}_2) - f(\mathbf{x}_1, \mathbf{y}_1))/\tau)) + \log(1 + \exp((f(\mathbf{x}_2, \mathbf{y}_1) - f(\mathbf{x}_1, \mathbf{y}_1))/\tau)) | \mathcal{E}^c] \\
&= (1 - 2p_2)^2(\mathbb{E}[\log(1 + \exp((f(\mathbf{x}_1, \mathbf{y}_2) - f(\mathbf{x}_1, \mathbf{y}_1))/\tau)) | \mathcal{E}] + \log 2) \\
&\quad + \Pr[\mathcal{E}^c]\mathbb{E}[\log(1 + \exp((f(\mathbf{x}_1, \mathbf{y}_2) - f(\mathbf{x}_1, \mathbf{y}_1))/\tau)) + \log(1 + \exp((f(\mathbf{x}_2, \mathbf{y}_1) - f(\mathbf{x}_1, \mathbf{y}_1))/\tau)) | \mathcal{E}^c] \\
&\leq (1 - 2p_2)^2(\mathbb{E}[\log(1 + \exp((f(\mathbf{x}_1, \mathbf{y}_2) - f(\mathbf{x}_1, \mathbf{y}_1))/\tau)) | \mathcal{E}] + \log 2) \\
&\quad + (4p_2 - p_2^2)(2\log(1 + 2/\tau)) \\
&\leq (1 - 2p_2)^2(\mathbb{E}[\log(1 + \exp((f(\mathbf{x}_1, \mathbf{y}_2) - f(\mathbf{x}_1, \mathbf{y}_1))/\tau)) | \mathcal{E}] + \log 2) + 24p_2/\tau
\end{aligned}
$$

where the third equality comes from the fact that $\mathbf{x}_1 = \mathbf{x}_2$ when $\mathcal{E}$ happens.

Since $f_0(1, \theta_0) = f_0(1, -\theta_0)$, we have

$$
\begin{aligned}
L_{\mathcal{D}^B}(f_0, \tau) &\leq (1 - 2p_2)^2(\mathbb{E}[\log(1 + \exp((f(\mathbf{x}_1, \mathbf{y}_2) - f(\mathbf{x}_1, \mathbf{y}_1))/\tau)) | \mathcal{E}] + \log 2) + 24p_2/\tau \\
&= 2(1 - 2p_2)^2 \log 2 + 24p_2/\tau
\end{aligned}
$$

For any $\theta \in [-\pi + \theta_0, -\theta_0] \cup [\theta_0, \pi - \theta_0]$, we have $|\cos(\theta + \theta_0) - \cos(\theta - \theta_0)| \geq 1 - \cos(2\theta_0)$. So, we have

$$
\begin{aligned}
\mathbb{E}\left[\log(1 + \exp((f_\theta(\mathbf{x}_1, \mathbf{y}_2) - f_\theta(\mathbf{x}_1, \mathbf{y}_1))/\tau)) | \mathcal{E}\right] &= \frac{1}{2}\log 2 + \frac{1}{4}\log(1 + \exp((f_\theta(1, \theta_0) - f_\theta(1, -\theta_0))/\tau)) \\
&\quad + \frac{1}{4}\log(1 + \exp(-(f_\theta(1, \theta_0) - f_\theta(1, -\theta_0))/\tau)) \\
&\geq \frac{1}{2}\log 2 + \frac{1}{4}\log(1 + \exp((1 - \cos(2\theta_0))/\tau)) \\
&\quad + \frac{1}{4}\log(1 + \exp(-(1 - \cos(2\theta_0))/\tau))
\end{aligned}
$$

And

$$L_{\mathcal{D}^B}(f_\theta, \tau) \geq (1-2p_2)^2(\mathbb{E}[\log(1 + \exp((f_\theta(\mathbf{x}_1, \mathbf{y}_2) - f_\theta(\mathbf{x}_1, \mathbf{y}_1))/\tau))\,|\,\mathcal{E}] + \log 2)$$

$$\geq (1-2p_2)^2(\frac{3}{2}\log 2 + \frac{1}{4}(\log(1 + \exp((1 - \cos(2\theta_0))/\tau) + \log(1 + \exp(-(1 - \cos(2\theta_0))/\tau))))$$

So, we have

$$L_{\mathcal{D}^B}(f_\theta, \tau) - L_{\mathcal{D}^B}(f_0, \tau) \geq (1-2p_2)^2(\frac{3}{2}\log 2 + \frac{1}{4}(\log(1 + \exp((1 - \cos(2\theta_0))/\tau) + \log(1 + \exp(-(1 - \cos(2\theta_0))/\tau))))$$

$$- \left(2(1-2p_2)^2\log 2 + 24p_2/\tau\right)$$

$$= (1-2p_2)^2(-\frac{1}{2}\log 2 + \frac{1}{4}(\log(1 + \exp((1 - \cos(2\theta_0))/\tau) + \log(1 + \exp(-(1 - \cos(2\theta_0))/\tau))))$$

$$- 24p_2/\tau$$

$$\geq \frac{1}{4}F(\theta_0, 1/\tau) - 24p_2/\tau$$

where $F(\theta_0, 1/\tau) = -\frac{1}{2}\log 2 + \frac{1}{4}(\log(1 + \exp((1 - \cos(2\theta_0))/\tau) + \log(1 + \exp(-(1 - \cos(2\theta_0))/\tau)))$.

Since $F(\theta_0, 1/\tau) > 0$ and $F(\theta_0, 1/\tau) = \Theta(1/\tau)$ when $\tau \to 0$, there is a constant $C$ such that $F(\theta_0, 1/\tau) < C/\tau$. Let $C' = C/8$ and $\delta' = C/192$, we complete the proof. □

## B  DETAILS OF EXPERIMENTS

We use the DataComp-small as the training set, released by DataComp (`https://www.datacomp.ai/`) under Creative Common CC-BY-4.0 license. We develop based on the codebase in `https://github.com/mlfoundations/datacomp`, licensed under MIT License.

## C  ADDITIONAL RESULTS

Figure 4 and Figure 5 show the results of our models on commonly used CLIP benchmarks.

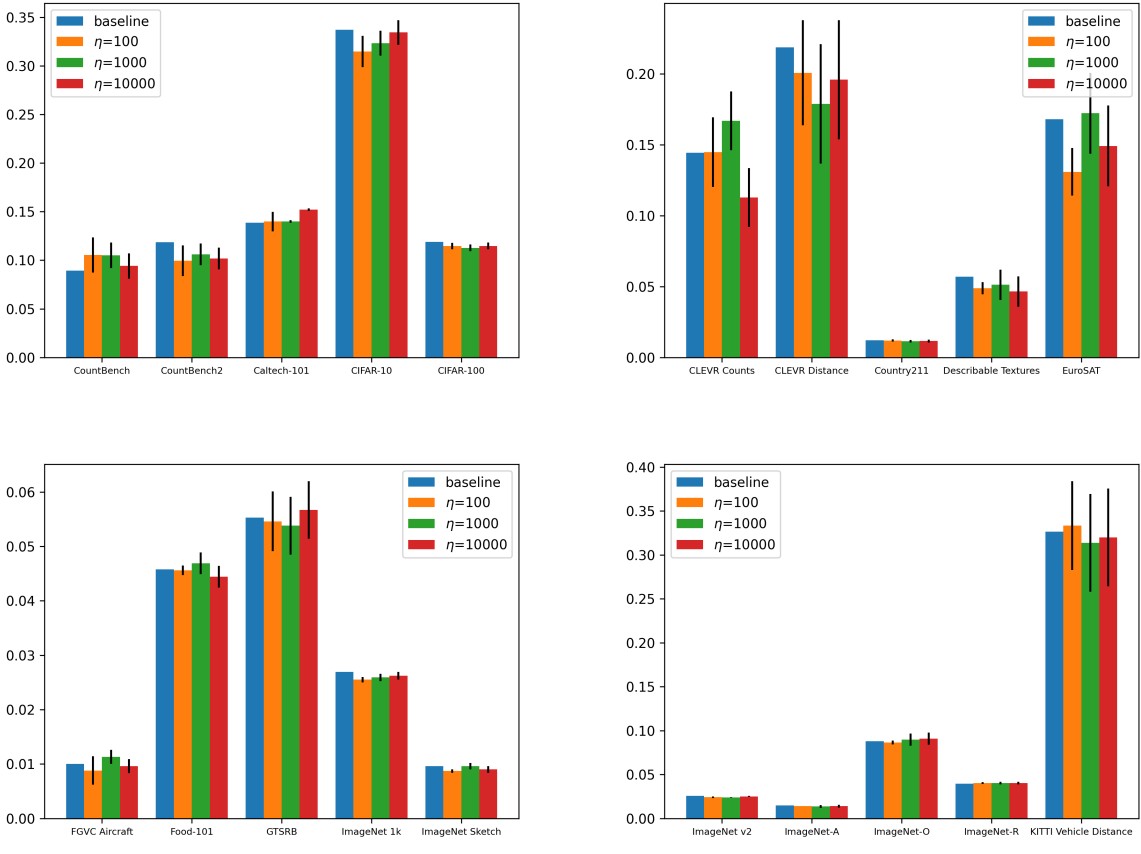

Figure 4: Results of other datasets with $g(x) = \log(1 + x)$.

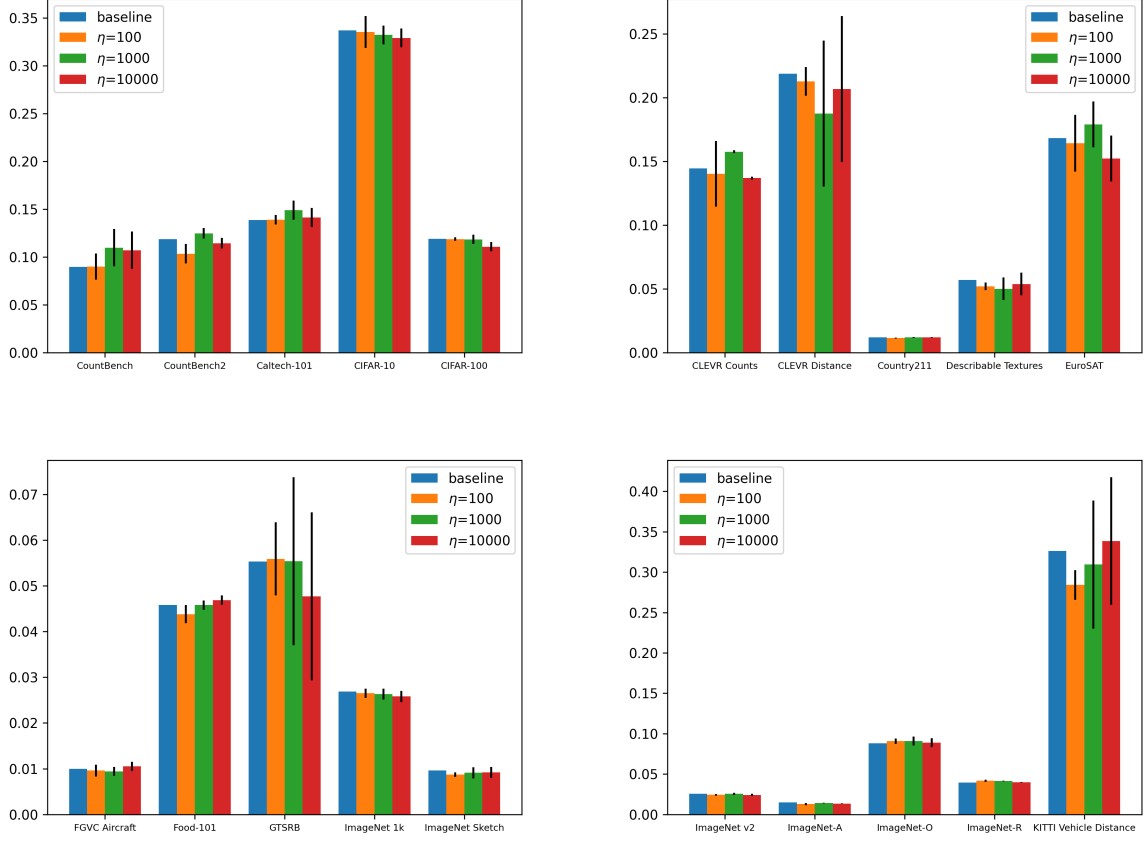

Figure 5: Results of other datasets with $g(x) = \frac{x}{1+x}$.

