# OpenReview forum: "Improved CLIP Training Objective on Fine-Grained Tasks: Tackling False Negatives and Data Noise"
_TMLR — Rejected by TMLR_

### Review · Reviewer_Aqou · 2025-10-09

**Summary Of Contributions:**

The paper proposes a modified CLIP training objective to improve performance on fine-grained image-text matching tasks (e.g. counting, spatial reasoning) by tackling false negatives and data noise during contrastive pretraining. The core idea is to add an auxiliary supervised contrastive loss that emphasizes true negatives, pairs known to have different semantic labels, and to modify the InfoNCE loss using a concave increasing function g(x) to make the training more robust to mislabeled data. The authors provide theoretical analysis proving that their objective can achieve bounded top-r error under weaker assumptions than prior work and show empirical improvements on the CLEVR-Count benchmark using DataComp-small.

**Audience:**

Yes

**Audience Explanation:**

Researchers working on theoretical analysis of contrastive learning or fine-grained multimodal training would find the paper interesting, especially its relaxation of CLIP’s prior assumptions and its theoretical bounds.

**Claims And Evidence:**

Yes

**Claims Explanation:**

Partially supported.

The theoretical analysis is sound and internally consistent, but the empirical evidence is limited and lacks breadth. The claimed robustness to noise and fine-grained improvement is demonstrated only in a toy setting (counting on CLEVR-Count) with modest gains. The paper’s central assumption that supervised true-negative labels can be easily obtained is not substantiated with clear methodology, datasets, or reproducible details. Consequently, while the mathematical claims are convincing, the practical claims are weakly supported.

**Requested Changes:**

The whole method of the paper is based on the assumption that we somehow have the supervised labels for "true negatives", and then based on this assumption to develop the theorems and the modified loss function. However, the author didn't explain much how to obtain such supervised labels. The most relevant text I can find is:

> To obtain the supervised signal, we use English cardinal numbers from “two” to “twenty” as the keywords and perform keyword matching on captions.
> We note that the keyword matching strategy can be further applied to obtaining similar fine-grained supervised labels like directions and colors.

This is so vague and thus it doesn't meet my standard to be an important part of the proposed "method" in a paper. You should be able to write an algorithm in pseudo-code or math equations. What's the set of English cardinal numbers? What's the set of the keywords you're using? (I don't believe the set only contains two and twenty.) What's the set of directions and colors? How do you do the string matching? We need these details if it's the core parts of the proposed method, which I believe it is.

Also, for the experiments, it's tested on very small-scale and narrow settings: ViT-B/32 small model, small batch size of 4K, narrow evaluation of CLEVR-Count. I saw some more regular CLIP evaluation benchmark results in appendix like ImageNet, but the accuracy of it is very poor due to the scale (under 10%), which could not be a good evidence. Also the trend is not consistent on all the benchmarks.

The core motivation of CLIP paper is to learn from the noisy signals in the raw text-image pairs from web data, and the scaling (large batch/large data/model size) helps the model reduce the noisy by itself. The trick of considering signals from keyword matching (numbers/directions/colors) may be helping in a small scale in specific benchmarks, but when we scale the batch size/data size/model size, the benefits may be marginal eventually. I wonder if the author can provide some evidence of the scalability of this method? Also, the collection of keywords need additional human efforts. It may be trivial to obtain such labels for English, but how about text image pairs not in English? It's also possible that these kind of small smart tricks can only be the obstacles during scaling at the end, and the gains from the smart tricks becomes marginal after scaling.

---

> ### Author Response · Authors · 2025-10-30
>
> We are grateful to the reviewer for providing comments on our submission!
>
> 1. regarding how to obtain the supervised labels, what we do is to use the English keywords "two", "three", "four", "five", ... until "twenty", and check whether one of the words in the caption sentence is a keyword.
>
> 2. regarding the experiments, we acknowledge that the experiment part is not fully prepared, and testing the results in a larger scale data will be more convincing. We will strive to improve these experiments.

---

### Review · Reviewer_VPeX · 2025-10-09

**Summary Of Contributions:**

The authors observe that CLIP’s standard contrastive loss mislabels many visually similar image–text pairs as negatives and is sensitive to noisy, web-crawled captions, which hurts performance on fine-grained tasks such as counting. They add a lightweight supervised signal—obtained cheaply by keyword matching (e.g., numbers in captions)—and introduce an extra loss term that only contrasts samples whose labels differ, guaranteeing they are true negatives. They also replace the usual log(1+x) form of InfoNCE with alternative concave functions that down-weight outliers, providing a tighter theoretical bound on top-r retrieval error under weaker distributional assumptions than prior analyses. Experiments on the CLEVR-Count benchmark with the DataComp-small pre-training set show consistent gains over baseline CLIP (e.g., ~17 % vs. 14 % accuracy) without additional annotated data, while an ablation that turns false negatives into positives is less effective. Overall, the paper claims its loss is simple to implement, theoretically justified, and readily transferable to other fine-grained multimodal tasks.

**Audience:**

Yes

**Audience Explanation:**

The findings of this paper would be of interest to a significant portion of the TMLR audience. The work addresses the challenge of improving the fine-grained understanding of CLIP, which is a highly relevant and important problem in the current machine learning landscape.

**Broader Impact Concerns:**

I have no broader impact concerns to report for this work.

**Claims And Evidence:**

No

**Claims Explanation:**

1. The paper's central premise is that an algorithmic modification to the loss function is the key to improving fine-grained understanding. However, an alternative and perhaps more direct approach would be to address the problem at its source: the data itself. Since the limited quality of web-crawled image-text pairs is a well-known bottleneck, a crucial question is whether the proposed method is a fundamental improvement or a workaround for poor data quality. A more compelling contribution could involve a data-centric pipeline that synthesizes or collects high-quality, detailed image-text pairs. To better position the current work, it would be important to conduct an experiment where the proposed loss function is compared against the baseline CLIP loss when both are trained on an improved dataset (e.g., one with captions generated by a modern VLM). This would help determine if the proposed method provides benefits beyond simply mitigating noise in the original, low-quality training data.

2. The empirical validation is narrow and may not be sufficient to support the paper's broader claims about improving fine-grained understanding. The experiments are confined to a single counting task (CLEVR-Count), which limits the generalizability of the findings. To demonstrate robust fine-grained capabilities, the evaluation should be extended to other important and diverse benchmarks, such as fine-grained classification (e.g., iNaturalist 2017 and CUB). Furthermore, since CLIP models are foundational vision encoders for many large multimodal models (LMMs) like LLaVA, a compelling way to demonstrate real-world impact would be to substitute the standard CLIP encoder with the authors' proposed model and evaluate performance on established multimodal benchmarks (e.g., MME, MMBench, MMVP). Finally, the current experiments only compare against the baseline CLIP, lacking comparisons to other relevant methods that also aim to tackle false negatives or improve fine-grained performance.

3. The presentation and visual design of the figures could be significantly improved for clarity and space efficiency.

**Requested Changes:**

My requested changes are based directly on the weaknesses identified in my review. Addressing them would substantially improve the paper.

---

> ### Author Response · Authors · 2025-10-30
>
> We appreciate the reviewer's effort in reading our submission and offering helpful feedback!
>
> 1. Regarding the visual design, we changed the position of some pictures to improve clarity.
> 2. Regarding the experiment, we acknowledge that the experiment part is not fully prepared and we agree that providing results on additional benchmarks and better datasets, and comparing them with other baselines would be more convincing. will strive to include these in our experiments in the future.

---

### Review · Reviewer_EjZT · 2025-10-18

**Summary Of Contributions:**

This paper identifies two key challenges that limit CLIP's performance: "false negatives" in the contrastive loss and the noisy training data. To address these problems, the authors propose a novel training objective. First, they introduce a new loss term that uses supervised signals to explicitly emphasize true negative pairs. Second, they modify the InfoNCE loss to be more robust to data noise. The authors provide a theoretical analysis to support their method, demonstrating its effectiveness. They validate their approach empirically on the CLEVR-Count benchmark, showing improved performance over a baseline CLIP model.

**Audience:**

Yes

**Audience Explanation:**

The primary strength of this paper is its novel approach to improving contrastive training. And the paper is well-supported by a theoretical analysis that demonstrates how the proposed method can relax some of the restrictive data assumptions made in previous analyses of CLIP.

**Claims And Evidence:**

No

**Claims Explanation:**

1. The most significant weakness of this work is the insufficiency of its experimental validation. The authors only provide a single benchmark (CLEVR-Count). While the authors claim their method is flexible and can be extended, this claim has not been justified with empirical evidence.
2. The authors provided a limited comparison against existing methods. The paper only compares its results against a baseline CLIP model. There is a notable lack of comparison to other established methods designed to improve CLIP's fine-grained understanding. This makes it impossible to assess whether it offers a competitive advantage over the current state-of-the-art.
3. Based on the limited experiments presented, the improvement over the baseline is very modest. For example, in Table 1, the top-1 accuracy on CLEVR-Count improves from 14.05% for the baseline to 16.69% for the best-performing variant.

**Requested Changes:**

1. Evaluation on a diverse set of fine-grained tasks beyond counting.
2. A comprehensive comparison with relevant state-of-the-art methods.
3. A deeper analysis of the results to better understand when and why the proposed loss function is effective.

---

> ### Author Response · Authors · 2025-10-30
>
> We appreciate the reviewer's effort in taking time to read our submission and provide constructive suggestions!
>
> Regarding the experiments, we thank the reviewer for pointing out the insufficiency of fine-grained tasks and the limited comparison against existing methods. We will strive to improve these experiments.

---

### Decision · Action_Editor_ijTe · 2025-11-29

**Recommendation:** Reject

**Audience:**

Yes

**Audience Explanation:**

CLIP image-text classification is of interest to the TMLR audience.

**Claims And Evidence:**

No

**Claims Explanation:**

This paper identifies two key challenges that limit CLIP's performance: "false negatives" in the contrastive loss and the noisy training data. To address these problems, the authors proposed a novel training objective. First, they introduce a new loss term that uses supervised signals to explicitly emphasize true negative pairs. Second, they modify the InfoNCE loss to be more robust to data noise. The authors provide a theoretical analysis to support their method, demonstrating its effectiveness. They validate their approach empirically on the CLEVR-Count benchmark, showing improved performance over a baseline CLIP model.

The three reviewers' considered the experiments to be insufficient and inadequate to validate the authors' claims:
-The manuscript only provides a single benchmark (CLEVR-Count). While the authors claim their method is flexible and can be extended, this claim has not been justified with empirical evidence.
-The manuscript provided a limited comparison against existing methods. The paper only compares its results against a baseline CLIP model. There is a notable lack of comparison to other improved versions of CLIP.
-Another reasonable experiment proposed by the reviewer is replacing the existing CLIP encoder of the state-of-the-art VLM by the proposed one to demonstrate the competitiveness of the proposed approach.

The authors' rebuttal is too simple and didn't solve any of the reviewers' concerns. The authors themselves also "acknowledge that the experiment part is not fully prepared" but didn't include any new experiments during the rebuttal. All of the reviewers recommended rejection of this manuscript.

**Resubmission Of Major Revision:**

The authors may consider submitting a major revision at a later time.